# Mixed Fault Classification of Sensorless PMSM Drive in Dynamic Operations Based on External Stray Flux Sensors

**DOI:** 10.3390/s22031216

**Published:** 2022-02-05

**Authors:** Sveinung Attestog, Jagath Sri Lal Senanayaka, Huynh Van Khang, Kjell G. Robbersmyr

**Affiliations:** Department of Engineering Sciences, University of Agder, Jon Lilletunsvei 9, 4879 Grimstad, Norway; sveinung.attestog@uia.no (S.A.); jagath.senanayaka@uia.no (J.S.L.S.); huynh.khang@uia.no (H.V.K.)

**Keywords:** demagnetisation, inter-turn short circuit, machine learning, permanent magnet synchronous motor, variable speed, variable load

## Abstract

This paper aims to classify local demagnetisation and inter-turn short-circuit (ITSC) on position sensorless permanent magnet synchronous motors (PMSM) in transient states based on external stray flux and learning classifier. Within the framework, four supervised machine learning tools were tested: ensemble decision tree (EDT), k-nearest neighbours (KNN), support vector machine (SVM), and feedforward neural network (FNN). All algorithms are trained on datasets from one operational profile but tested on other different operation profiles. Their input features or spectrograms are computed from resampled time-series data based on the estimated position of the rotor from one stray flux sensor through an optimisation problem. This eliminates the need for the position sensors, allowing for the fault classification of sensorless PMSM drives using only two external stray flux sensors alone. Both SVM and FNN algorithms could identify a single fault of the magnet defect with an accuracy higher than 95% in transient states. For mixed faults, the FNN-based algorithm could identify ITSC in parallel-strands stator winding and local partial demagnetisation with an accuracy of 87.1%.

## 1. Introduction

Fault diagnosis in permanent magnet synchronous motors (PMSM) has always received a lot of attention, aiming at increasing the safety level in dynamic and critical applications or preventing large economical losses due to unexpected downtime [1]. Unlike industrial productions, PMSMs in wind turbines and electric vehicles are intensively exposed to dust, mechanical, chemical, and thermal stresses in harsh environments and thermal cycling due to the dynamic operations. This causes an inter-turn short-circuit (ITSC) in the stator windings and demagnetisation on rotor permanent magnets (PM) of PMSMs, resulting in reduced efficiency and increased cogging torque [2]. The ITSC is a common fault in all motor types, e.g., induction motor and reluctance motor. However, demagnetisation is a fault that is unique to permanent magnet motors. A uniform demagnetisation affects all magnets equally and significantly reduces the overall back electromagnetic motive force (BEMF). A local demagnetisation only affects a specific region of rotor magnets. The complexity of machine structure, variable speeds, and loads render fault diagnosis challenges due to sensor requirements and nonstationary signals, attracting significant research that has tackled ITSC and demagnetisation in industry and academia in recent years, which is briefly discussed hereafter.

### 1.1. Related Works

Most research on ITSC implements the short-circuit in a single-strand stator winding. The ITSC in such windings can be detected based on characteristic frequencies in the current measurement. However, stator windings in PMSM can have multiple strands in parallel. Such stator windings have lower stator resistance and inductance, but the maximum output of the BEMF is reduced. The impact of ITSC in multi-strand windings on current signals is less compared to its single-strand counterpart. Multiple strands in one phase with an ITSC may be unaffected by it. The fault signature will be different if the ITSC is on a single stand or between two strands. Van Der Geest et al. [3] have numerically analysed a PMSM with three parallel strands in the stator to study the effect of ITSC on the copper losses. The authors proposed a fault detector, which was the fundamental component of the difference between machine neutral voltage and the estimated inverter neutral voltage based on three equal impedance. The method has been tested on both a high-speed PMSM prototype and an off-the-shelf PMSM. It was not possible to implement ITSC on the prototype; therefore, two additional turns were wound around two stator teeth to resemble an ITSC. The research does check the performance of the fault detector in case of "open strand fault", partial demagnetisation, and misalignment in simulation. It was found that the fault indicator was most sensitive to ITSC fault and could detect it with a severity level down to 0.4%. However, it is not clearly specified if the ITSC is applied to all strands or just a single strand. Furthermore, it is unclear whether the extra turns were added to a single strand or to the end terminal. The fault indicator was tested in steady state operation, but at different speeds. The loads in their experiments were either no-load or at the rated condition. No variable speed or load profiles were used to test the detector.

Zhu et al. [4] investigated a partial demagnetisation fault, which was assumed to be applied uniformly on all magnets. The rotor was demagnetised by increasing the internal temperature by overloading. The current level of demagnetisation was estimated based on the measured BEMF in two study cases, where the magnets had lost 17% and 33% of their original strength. The merit of this method is that it does not cause any physical damage or imbalance in the rotor, but the demagnetisation is implemented uniformly. The Vold–Kalman filtering for an order tracking was used to detect demagnetisation alone in a PMSM. Kao et al. [5] proposed a diagnosis algorithm, combining discrete wavelet transform and convolution neural network (CNN), to separate between local demagnetisation and bearing faults. Two strategies for demagnetisation were implemented: first replacing a piece of the magnet with aluminium, and learning how to weaken the strength of the magnet. The paper does not address the second method: how to weaken the magnet. However, validating this approach for detecting demagnetisation is not convincing, since there is an imbalance if the replacement material does not have the same density. The fault classifier proposed in [5] is based on one-dimensional CNN, being trained for demagnetisation fault and bearing fault, but the authors did not investigate the possibility for mixed fault context. The classifier was trained and tested for 20 different constant speed settings, but there is no variable speed profile in this work.

Iglesias-Martinez et. al. [6] investigated the use of bi-coherence and a fuzzy C-means algorithm for detecting winding asymmetries in induction motors. External coil sensors with 1000 turns and an external diameter of 8 cm were placed in proximity to the motor, measuring radial, tangential, and axial components of the stray flux. The stray flux signals were recorded during the start-up of the motor; then, they were used in a fuzzy C-means machine learning fault classifier, resulting in an accuracy of about 90%. Zamudio-Ramirez et. al. [7] proposed also smart sensors for the online detection of individual and combined faults in induction motors, such as broken rotor bar and misalignment. The sensor solution consists of three hall sensors oriented towards the three Cartesian directions. Their analysis concluded that the axial and radial stray flux components were the most effective in the transient analysis. The classification was performed with linear discriminant analysis and feedforward neural network (FNN), where the time-frequency representations were computed from the MUSIC algorithm during startup. The classifiers were trained by data from different mechanical faults such as broken rotor bar and misalignment, but mixed faults were missing in the study. The neural network was trained on measurements from startup duration alone, without other operations.

Gurusamy et al. [8] conducted a theoretical study on how loading would affect the fundamental and third harmonic of the measured stray flux based on finite element analysis. The information of stray flux was extracted from selected points in the finite-element analysis. The results showcased how the harmonics would change in magnitude, depending on the location of the fault. Simulations were verified with experimental work. The third harmonic of the stray flux was proposed as the fault indicator, since it was less affected by loading conditions than the fundamental component. However, the third harmonic may indicate many kinds of asymmetries in the magnetic circuit. Furthermore, the ITSC detector was only tested for constant speed or load operations.

### 1.2. Contribution

Conventional methods of detecting faults in stator winding and PMs of PMSMs methods focus on off-line tests, steady-state operations, and require a high number of available sensors in the drives. However, PMSMs are always controlled and dynamically operated based on the driver’s demands. Furthermore, modifying the existing drive system for automatic fault detection must be avoided in critical systems and applications, where sensorless controls are preferred due to reliability or reduced maintenance cost. Stray flux has recently gained popularity in the literature for fault detection of ITSC, demagnetisation, and mechanical faults due to its sensitivity and is simple to install outside of a drive system. Existing research focuses on model or signal-based algorithms, requiring multiple stray flux sensors and speed measurements for detecting faults in variable speed conditions. Furthermore, seeded demagnetised magnets were inappropriately implemented in the existing research, causing imbalances or without an overheat demagnetisation. To the author’s knowledge, most existing machine learning-based fault classifications are trained and tested on the same datasets or the same operational profiles, rendering a challenge of trustworthiness on other datasets or working conditions. To address the mentioned challenges, this paper aims to achieve the following:Increase the robustness of potential machine learning classifiers against transient operating conditions and different operation profiles by resampling at a fixed angular increment.Eliminate the need of position sensors in the resampling process (order tracking).Introduce a heat treatment method for inducing local partial demagnetisation fault.Train the machine learning algorithms, ensemble decision tree (EDT), k-nearest neighbour (KNN), support vector machine (SVM), and feedforward neural network (FNN) and compare their performance in different operation profiles and with or without the presence of mixed faults in the datasets.Investigate the variation of the classifier accuracy based on features computed from data collected by stray flux, current, and torque signals.Recommend minimum length of data sample for an accurate fault classification.

The rest of this paper is organised as follows: Section 2 describes the proposed technique for fault diagnosis under stationary as well as nonstationary conditions, which was followed by a description of how the fault was implemented in Section 3. Section 4 describes the experimental setup of the in-house test bench. The results are presented and discussed in Section 5. This paper ends with concluding remarks in Section 6.

## 2. Methodology

The block diagram in Figure 1 shows the proposed pre-treatment of measurements for computing features before passing it through a machine learning classifier. First, data samples were collected and labelled based on their fault case and operation condition. The rotor position was estimated with an optimisation problem. The estimated position was used for resampling of the original datasets, with data points sampled at a fixed time increment, to datasets sampled at a fixed angular increment. The resampled datasets were split into smaller samples, which contain a fixed number of rotor revolutions. The second step is to compute the spectrograms of the resampled datasets with Fast Fourier Transform (FFT), which is normalised based on the fundamental frequency. This gives a fundamental frequency spike with an order number equal to 1 and a magnitude of 1. Therefore, any fault classification approach based on analysing these spectrograms needs to study the composition of the harmonics. Third, an envelope of the spectrogram was obtained by splitting the spectrogram into intervals with a length of 0.5 order centred around the half harmonics [0.5, 1, 1.5, …, 39.5, 40] and then finding the maximum in each interval. This simple method produces an envelope of the spectrogram, which includes the harmonic peaks with fewer entries. In addition, the pre-treatment will order normalise the spectrogram and make classification more robust against speed change. The resulting envelopes of the spectrograms are chosen as the selected features for classifications, being trained and tested on EDT, KNN, SVM, and FNN.

### 2.1. Resampling Time-Series Data

Measured signals in PMSM, e.g., flux, current, and voltage are cyclical and ideally have a sinusoidal shape. In the case of a four-pole motor, every two cycles in these measurements represent one revolution of the rotor. Normally, datasets are sampled at constant time intervals. There will be more data points per cycle at lower speeds. For example, there are 400 data points per cycle if a four-pole motor is running at 750 rpm with a sampling frequency of 10 kHz. The number decreases to 100 data points per cycle when the motor is running at 3000 rpm. This imbalance of samples per cycle is the main cause for the peaks spreading over a wide range of frequencies in the spectrogram of signals with time-varying frequencies, which is computed by FFT. A solution to this is to resample the data from equal time steps to equal angular steps [9].

Figure 2 shows the results of the resampling process with a chirp function. The blue rings represent samples from the original dataset, where the data points are spaced at an equal time distance. Figure 2 highlights that more samples are concentrated at the first interval, when the instantaneous frequency is at its lowest. The red points represent the resampled dataset with the equal angular distance, where every interval have the same number of samples regardless of frequency. The resampling process can be viewed as a transformation from time-domain to θ-domain. After the transformation, the datasets can be split into intervals defined by a fixed number of cycles instead of a fixed time interval. Longer samples, regardless of being defined by time intervals or cycles, will increase the order resolution in the spectrograms and improve the confidence in the present peaks. The minimum number of cycles needed for computing features will be discussed in the results.

### 2.2. Estimation of Position

The resampling process requires the information of the rotor position or position sensor [10]. However, being dependent on encoders, resolvers, or tachometers will exclude the applicability in sensorless PMSM drives, since these sensors are not available. Removing a position sensor will improve the system reliability and enhance the performance of any fault classification method. Etien et al. [11] proposed a nonlinear adaptive algorithm with phase-locked loop and orthogonal signal generators. This method is suited for online detection but does require tuning a PI controller. Lu et al. [12] computed the phase from a filtered current signal, which is turned into a unipolar signal by an absolute function. The signal is divided into frames, which is every half cycle of the sine function. The phase angle is computed from the inverse sine of the ratio between the instantaneous value of the function within the current frame and the maximum value of the previous frame. This method is more prone to noise, because of the few key data points defining the maximum value of each frame. The method suggested in this paper is to find the fundamental frequency via an optimisation problem. Hou et al. [13] proposed ridge regression for obtaining instantaneous frequencies in time-frequency representation. Then, this method requires wavelet or short-time Fourier transform. The suggested method does not need such algorithms for estimating the fundamental frequency. Assuming the speed changes is negligible in a short time period (0.1 s is used in this study), the fundamental component of the signal can be defined as follows:(1)$$h\left(t\right)=\sin\left(2\pi f_{1}t+\theta \right)$$
where $f_{1}$ is the fundamental frequency and θ is the phase angle. The fundamental frequency can be obtained via FFT, but short-time periods are needed for an accurate estimate, which negatively affects the frequency resolution. Instead, the fundamental frequency is estimated by the optimisation problem:(2)$$\left[f_{1},\theta \right]=\underset{f_{1},\theta }{arg min}\left({\left(x(t)-h(t)\right)}^{2}\right).$$

The range for the optimal value of $f_{1}$ can be narrowed down between the standstill and nominal speed, i.e., $f_{1}\in \left[0,100\right]$ Hz. The value of the objective function in (Equation 2) repeats itself when θ is increased from 0 to 2π, keeping the $f_{1}$ constant. The modulo operator is used such that θ∈[0,2π]. Steps to estimate the fundamental component in the stray flux signal are described as follows.

Extract a small time sample with period T from the original time-series data.Compute the objective function for frequencies in the interval [0,100] Hz with an incremental step of 5 Hz. The optimal ϕ for a given $f_{1}$ is found by the Golden Section Search in the interval [0,2π]. The values for $f_{1}$ and ϕ, that yield the smallest value of (Equation 2) are the initial guesses in step 3.Find the optimal solution for $f_{1}$ and ϕ by the Simplex Search method in [14] with the initial guesses given in step 2.Repeat step 1 to 3 for the next time step until the end of the time sample.

The output of the estimator is dependent on all data points in the sample of each iteration. Therefore, it would be less prone to noise and require no tuning. One weakness to this approach for obtaining the fundamental frequency is a local convergence. Signals with an immoderate signal-to-noise ratio will tend to reach 0 Hz instead of the desired solution. One approach to avoid this problem is to increase the lower boundary in the brute force analysis in step 2, e.g., 10 Hz.

### 2.3. Machine Learning Methods

This section aims to investigate the performance fault classifications of features computed from stray flux measurements. The following supervised machine learning algorithms are investigated: DT, KNN SVM, and FNN. Each sensor produces an envelope of the spectrogram with 41 elements; thus, the number of inputs is the product between 41 and the number of sensor measurements added to compute the features.

#### 2.3.1. Ensemble Decision Tree

The “trees” in decision tree classification starts at the first node known as the trunk, then it branches out to more if-else statements. The final nodes of the tree are called leaves, which get numeric values in the case of regression tree and discrete categories in case of classification tree. An EDT consists of a forest with smaller decision tress called "stumps". They usually only have one node with two leafs each. The “stumps” are weak learners on their own, but combined ones can predict with high accuracy where the class is decided with a majority vote. However, the votes from each “stump” got a weight, which is obtained from training. The EDT in this study uses the AdaBoost method for training with 100 training cycles with the learning rate of 1. The maximum number of splits per tree is set at 1, and each leaf has a minimum of 1 observation [15].

#### 2.3.2. K-Nearest Neighbours

KNN is one of the simplest supervised machine learning algorithms, but it is still an effective tool. The principle of the classification process is that new observations are categorised based on the previous observations closest to them. The number of neighbours, k, is used to decide the class of the new observation. It is adjusted to make the classifier more robust and less likely to overfit. The votes from the closest neighbours are counted, and the class with the highest number of votes dictates the outcome of the classifier [16]. In this paper, k was set to 15, which was obtained via trial-and-error during training. The distance is computed with Euclidean distance, and the potential ties in four-class classification are settled by the label of the nearest neighbour among the tied groups. Ties are not possible in fault detection (two-class classification), since k is an odd number. Each of the features is standardised by centring and scaling with respect to their mean and standard deviation.

#### 2.3.3. Support Vector Machine

In an SVM, the input data are mapped into a high-dimensional feature space using a kernel function. The objective is to maximise the margin between the classes and give the best separation of the training data [17]. In this paper, the linear kernel function was used. The nature of SVM is to classify between two datasets. These methods are called “one-versus-all” and “one-versus-one”. The first method uses multiple SVMs where each one determines if an observation belongs to their respective class. Ideally, only one SVM will identify the observation as their own class, which dictates the outcome of the classifier. The second type of multi-class SVM was used in this study. Multiple SVMs are trained to separate between two classes, which is unique to each SVM. A decision is reached with a plurality vote. Each of the features is standardised by centring and scaling with respect to their mean and standard deviation.

#### 2.3.4. Feedforward Neural Network

The mathematical model [7] for the output y in each neuron in an FNN is given as
(3)$$y=f\left(\sum_{n=1}^{N}w_{n}x_{n}+b\right)$$
where $w_{n}$ and *b* are the weights and bias applied to the inputs $x_{n}$, and f(·) is the activation function. The rectified linear unit (Relu) activation function was selected. A block diagram of the FNN used in this paper is shown in Figure 3, consisting of two fully connected layers. The first one contains 100 neurons, while the other is equal to the number of classes. In case of fault detection, it is 2, and it is 4 for classification with mixed faults. It also has a batch normalisation layer, which makes it faster and more stable. The last two layers are the softmax layer and the classification layer. The network is trained with an adaptive moment estimation [18] with a learning rate of 0.001, gradient decay of 0.9, squared gradient decay factor of 0.99, and denominator offset of $10^{-8}$. The mini-batch size and number of epochs are set to 16 and 10, respectively.

## 3. Implementation of Faults

### 3.1. Implementing Local Demagnetisation

Demagnetisation faults in permanent magnet motors are usually implemented by removing parts of the magnets [19] and replacing them with a non-magnetic material [20]. This is to counteract the imbalance in the rotor. Another option is to install weaker magnets in the rotor through manufacturers [21]. The mentioned implementations do not mimic the demagnetisation due to thermal cycling in dynamic operations of sensorless PMSM drives. This study presents a demagnetisation process by heat treatment. The temperature distribution on magnets is not uniform. The investigations performed by Fernandez et al. [22] and Reigosa et al. [23] showcased how the temperature is distributed under different applied currents in thermal equilibrium. The study was applied on PMSM with interior magnets, where they concluded that the hottest spot on average over time is in the middle of the magnet. Thus, this region is most likely to be affected by demagnetisation due to overloading.

Several attempts for local demagnetisation were executed involving an electric discharge machine, blow torch, and heat gun. None of the solutions can achieve a satisfactory result, where one magnet became partially demagnetised without causing significant physical damage to the rotor. Note that these methods were tested on a separate rotor that was not used in the final test. The best option, to our knowledge, is to heat one side of the rotor on a 1500 W electric cooking plate made of iron. A solid aluminium block with the dimensions of 200 mm × 90 mm × 20 mm was placed between the rotor and the cooking plate due to its low permeability. This ensures a safe placement and removal of the rotor on the hot cooking plate. A wet towel was placed on the rotor to cool the other poles. The heat treatment is shown in Figure 4.

The rotor was placed on the hot aluminium block, which reached temperatures up to 232 °C. The rotor was left on the hot surface for 5 minutes. The thermal paste was used to improve the heat transfer between the objects.

The magnetic field around the rotor was measured before and after the thermal treatment, as shown in Figure 5 with an Extech SDL900 magnetic field meter. Note that the measurements after the heat treatment were done after the rotor had cooled down to the ambient temperature. The wooden frame allowed the rotor to rotate, kept its rotor axis horizontal, and prevented any translational movement. The measured magnetic field strength surrounding the rotor decreases with distance. Therefore, the hall sensor in SDL900 was kept at a constant distance of 3 mm from the rotor surface.

Figure 6 shows the measured magnetic field surrounding the rotor before and after the heat treatment. The measurement was repeated three times to reduce the measurement error. The north poles are indicated by positive value with the south ones are negative. It can be seen that the rotor after the heat treatment has a slight decrease of magnetic strength at two points on one pole. Further investigations reveal that those two spots lost up to 30% of their original field strength.

### 3.2. Implementing Short Circuit Fault

In the studied PMSM, each phase winding has three strands in parallel. The advantage of this configuration is a reduced overall phase resistor and phase inductance. The drawback is a smaller BEMF. A simple analysis of the expected DC resistance measured by a multimeter was conducted. The inductance is ignored, since it does not affect the measured resistance in steady state.

It is assumed that the resistance of each strand is $R_{S}$. Four taps were implemented in one phase of the PMSM. The taps are shorted with the input voltage terminal U. This arrangement ensures that no short-circuit occurs between two parallel strands. Resistances between a tap and the end terminals were measured for each tap. A single multimeter can often measure resistance, but the accuracy is limited by the low current induced by the multimeter. Figure 7 shows the sensitive resistance measurement setup. A current source of 1 A is used, which is significantly less than the nominal current 6 A of the studied PMSM. The strand, where a tap is placed, is assumed to be split into two resistors $R_{1}$ and $R_{2}$. The measured equivalent resistance $R_{\mathrm{mea}}$ between a tap in the middle of the coil and the end or end terminals is defined as below.
(4)$$R_{\mathrm{mea}}=\mu_{f}\left(1-\frac{2}{3}\mu_{f}\right)R_{S},$$
where $\mu_{f}$ is the number of turns between the tap and the end terminals divided by the total number of turns in the coil. The solution for $\mu_{f}$ is the second-order equation:(5)$$\mu_{f}=\frac{3}{4}\pm \frac{1}{2}\sqrt{\frac{9}{4}-6\frac{R_{\mathrm{mea}}}{R_{S}}}.$$

The valid answer from (Equation 5) is identified based on the location of the tap relative to the end terminals. If the tap is expected to be close to the U-terminal, it is reasonable to assume that $\mu_{f}<\frac{3}{4}$. It is also reasonable to assume that $\mu_{f}>\frac{3}{4}$ between the tap and neutral. The ITSC severity of 5% was used in this study.

## 4. Experiment and Data Collection

### 4.1. In-House Test Bench

A schematic diagram of the test setup is shown in Figure 8, including a 400 V three-phase drive and the Microlabbox operated with an office laptop. The hall sensors measuring the stray flux were solid-state sensors of type SS495A. The output of these sensors are ratio-metric and linear within the range [−67, 67] mT and has a sensitivity of 31.25 $\frac{\mathrm{mV}}{\mathrm{mT}}$. Two sensors were soldered to a veroboard and wired to the Microlabbox, which delivered power to the sensors and recorded the stray flux. One of the sensors was bent 90°, such that it could measure both the tangential and radial component of the stray flux. Two sets of sensors were placed in proximity to the motor at the top and on the side. Furthermore, the current sensors are located inside a cabinet to the top right corner of Figure 9 to measure the input phase currents. The study motor has no position sensors; therefore, the encoder on the generator measure s the rotor position to validate the suggested method of position estimation.

Demagnetisation fault is introduced by replacing the rotor with no magnet defects with a rotor that went through the heat treatment process described in Section 3.1. Both rotors belong to the same type of PMSMs: IE5-PS2R 100 L, with their key parameters given in Table 1. ITSC faults are induced by wiring the external taps implemented in the PMSM. A fault resistor (1 Ω) is used for mimicking the remaining insulation in an ITSC and limiting the short circuit current. The motor is coupled to a generator with a torque transducer in between. Both the encoder and torque transducer are powered by a 24 V DC-supply. The output of the generator is rectified with a three-phase full-bridge rectifier. Two 1000 μF capacitors are connected in series across the output terminal of the rectifier, which removes the ripples of the DC output. The reason for two capacitors in series is that the output of the rectifier at nominal speed does exceed the voltage rating of the capacitor (400 V). The equivalent capacitance is 500 μF, but this is still sufficient to remove the voltage ripples. The brake chopper is regulated by a pulse width modulation (PWM ) signal from the Microlabbox, which needs to be amplified by a factor of 4 due to insufficient voltage amplitude from the Microlabbox. The op-amp is powered by a 12 V DC supply. In an ideal system, the duty cycle would be proportional to the reciprocal of the motor speed. However, due to losses and imperfections, a look-up table is generated for the duty cycle, which dictates the required duty cycle for achieving a requested load in the speed range between 1000 and 2000 rpm.

### 4.2. Description of Collected Datasets

Stray flux was measured for three different nonstationary operating conditions with a sampling rate of 10 kHz. The rotor position, input current, and torque were measured for all three profiles. Observation used for training and testing in the ML tools is computed from smaller samples. They are from three operation profiles, which are split based on a fixed number of revolutions of the rotor. The output of the resampling process obtained 400 samples per revolution. Section 5 will use the term “cycles”, which is double the number of revolutions of the rotor. Profile 1 contains a regular pattern, where the speed ramps up and down between 1000 and 2000 rpm. The load changes between 25% and 75% of the full load. The profile shown in Figure 10a includes combinations of increases and decreases of the load when the speed is increasing, decreasing, or constant. The second operation profile in Figure 10b keeps the reference speed constant at 1200 rpm, but the load changes with a pattern, which is randomly generated and repeats itself every 30 s. The last profile keeps the load steady at 60%, and the reference speed profile was also generated with random numbers. It is plotted in Figure 10c. The total period is 120 s for all three profiles. In the remainder of this paper, these profiles will be referred to as Profile 1, Profile 2, and Profile 3.

Data collection of all conditions of the PMSM was repeated for all mentioned profiles in the following fault condition: There were healthy conditions, ITSC with 5% severity, local partial demagnetisation, and a mixed faulty case of ITSC and demagnetisation.

## 5. Result and Discussion

### 5.1. Position Estimation

The purpose of estimating the rotor position is to remove the need for any position information or sensor in the fault classification and make the classifiers robust against speed changes. A fault classifier scheme using stray flux alone would be easier to implement in existing sensorless PMSM drives, since an external flux sensor can be placed in proximity of the PMSM. The optimisation problem, which estimated the speed, was able to estimate the electrical position $\theta_{e}$ from a flux signal with an average relative error of less than 0.5%. Therefore, the position measurements were replaced by the estimated position for the resampling process. The estimated electrical angle position and the measured value from the encoder are plotted in Figure 11.

### 5.2. Comparing Physical Parameters

The performance of the four learning classifiers, EDT, KNN, SVM, and FNN, is investigated. The recorded data of operation Profile 1 with regular change in speed and load is split in 80% for training and 20% for testing. The observation in Profile 1 is shuffled randomly before the split to prevent over-fitting of the classifiers. After training, all algorithms were tested on the whole datasets of Profile 2 and Profile 3, which were also shuffled, and the remaining observation from Profile 1. The performance of algorithms is varied depending on the training data. Therefore, the average achieved accuracy and training time are obtained by a Monte Carlo analysis, where the algorithm is repeatedly trained and tested 100 times. The training time is reported in Table 2. All the algorithms had a training time less than 3 s. However, the computation times of EDT and FNN of the first and second orders of magnitude are larger than the other two.

The letters “τ”, “I”, and “ϕ” in Figure 12, Figure 13 and Figure 14 refer to the use of torque, current, and stray flux signals for training the algorithms, respectively. The number refers to how many sensors were used to compute the features in each observation. For the torque signal, there is only one transducer coupled between the motor and generator. Three current sensors or one sensor per phase are used to collect current signals. There are four hall sensors in the setup, which can record the radial and tangential components in proximity of the motor through data acquisition. In the comparison study between physical parameters, only one pair of hall sensors is used.

For single faults, all four machine learning algorithms were trained on Profile 1 in the cases of only local demagnetisation or ITSC fault (severity 5%). The achieved accuracy of the classifiers is on the y-axes in Figure 12 and Figure 13 with a range between 50% and 100%. This highlights the differences in performance of the classifiers trained with different physical measurements and operation profiles. In case of ITSC fault, all classifiers obtain an accuracy less than 70% with the current and torque measurements. It appears that these physical parameters are not sensitive to ITSC. This phenomenon can be explained based on the configuration of the stator winding with three parallel strands. The short-circuit tap s are connected to the phase terminal, which is a common point for all three strands. The other end of the short-circuit was soldered to a single strand, making the remaining strands in parallel unaffected. Therefore, the 5% ITSC fault has less impact on the motor torque or total phase current, which is the sum of all the currents in the three strands. However, the hall sensors are significantly more sensitive to the presence of the ITSC fault. The algorithms EDT, SVM, and FNN can reach an average accuracy over 90% when testing on data of Profile 1. The purpose of the pre-treatment of the time-series datasets is to make the machine learning algorithms more robust against transient operating conditions. The strength of this process is further demonstrated by testing the machine learner on Profile 2 and Profile 3. None of the machine learners have been train with samples from these operation profiles. The accuracy of the ITSC detector drops from around 95% to around 85%, with SVM and FNN having the highest accuracy. The drop in accuracy implies that there is some over-fitting, and the machine learners can be confused by new unexpected operating conditions. However, the accuracy of 85% is still considered respectable. This is the accuracy when testing on one operation profile, which the machine learners have not been trained for, and an ITSC fault with 5% severity on one out of three parallel strands in a PMSM is less severe as compared to the case of one strand per phase. KNN has the lowest accuracy in this case study. It is hardly better than random chance when being trained on features computed from only torque or current data and tested on Profile 2 and Profile 3. Data from stray flux sensors on the other hand achieve an average accuracy of around 80%.

For the single fault of local demagnetisation, all four classifiers can reach an accuracy larger than 90% with features computed from stray fluxes, as shown Figure 13. It is noted that the data from stray flux sensors allows all four classifiers to detect the local demagnetisation with an accuracy of above 90%. The accuracy does not drop significantly when being tested on the unseen operation profiles (Profile 2 and Profile 3). This indicates that the asymmetry caused by local demagnetisation is detectable, and the classifiers become robust to unseen load and speed changes for identifying these signatures. The performance of the machine learners drops significantly if the current or torque data are used instead of computing the input features. The KNN-based classifiers have overall the worst accuracy among the tested algorithms. However, the accuracy achieved with features computed from stray flux is on par with the rest.

For the mixed fault, the machine learning algorithms are not only trained for fault detection but also for discrimination between local demagnetisation and ITSC. In this study, there are four classes: no-fault, only local demagnetisation fault, only ITSC fault, or mixed fault. There is an equal number of observations in each fault case; thus, the y-axes in all of the bar plots in Figure 14 are limited to above 25%. The EDT, SVM, and FNN-based classifier have the highest accuracy among the studied algorithms. The accuracy of all the classifiers can be increased by using more data for training, e.g., if all four hall sensors are included. The KNN -based classifier has an accuracy of just above 25% when being trained and tested using current or torque data on Profile 2 and Profile 3. In addition, the mixed fault does deteriorate the performance of the tested algorithms if they were trained on separate faults. This indicated that the fault signatures of local demagnetisation and ITSC in their incipient state are difficult to separate. However, the tested algorithms using stay flux signals can improve the performance in detecting these magnetic asymmetry.

All the average accuracies are plotted in Figure 12, Figure 13 and Figure 14 and reported in Table 3. The columns blow “Demagnetisation” and “ITSC” are in case of single fault. In case of demagnetisation, all algorithms achieve an accuracy of above 90%, which is highlighted in bold. The last three columns under the label "Mixed" refer to the mixed fault of demagnetisation and ITSC.

### 5.3. Required Samples for Fault Classification

Normally, the length of samples used as input for FFTs is measured in time. Since the time-series data have been resampled due to the transient operation condition, the length of each sample is defined by the number of cycles. Stray flux and current signals measured from a PMSM are cyclical due to their sinusoidal nature. One cycle is defined as one period of the fundamental frequency in the measurement. It is equal to double the number of rotor revolutions in the case of four-pole motors. The frequency or order resolution of the spectrogram computed by FFT depends on the length of the sample; thus, a longer sample would give more confidence in the presence of peaks. A longer sample will also equate to a longer measuring period, which depends on the motor speed. For example, 30 revolutions (60 cycles) of a four-pole PMSM running at 3000 rpm are equal to a measurement period of 0.6 s, but this will increase to 2.4 s when the motor is running at 750 rpm. Therefore, it is of interest to find the minimum number of cycles required for accurate fault detection.

Figure 15 shows line plots of the detection accuracy of local demagnetisation achieved by the FNN classifier, which has the highest accuracy at different numbers of cycles used to compute the spectrogram. The achieved accuracies were invested in cases when the features were being computed using data from one torque sensor (1 τ), three current sensors (3 I), two hall sensors (2 ϕ), and four hall sensors (4 ϕ). Combination of the three different physical parameters was also tested, but none of them gave a significant improvement as compared to the case of using four flux sensors. The accuracy in all four cases showcased in Figure 15 is increased if using data from more cycles. However, only the two cases using stray flux data do converge. The accuracy converge if using four-flux sensors data of 20 cycles and two hall sensors data of between 30 and 40 cycles. The FNN-based classifier takes longer to converge, implying that unseen speed changes may still offer a challenge in training machine learning classifier. It is recommended that one pair of flux sensors of at least 20 cycles, measuring tangential and radial components, should be used for a reliable identification of magnet defects. If the data have a length of 20 cycles, i.e., 10 rotor revolutions of the rotor, then the required measurement time for computing one set of features is 0.8 s when the motor operates at 750 rpm or 0.2 s when it works at 3000 rpm. The fault classifiers will perform better if being trained on features computed from longer datasets, but they will take longer to acquire the first sample. The required computing time of the next dataset of features can be significantly reduced if the dataset for the next set of features is overlapped with the previous one.

## 6. Conclusions

This paper presents a scheme of fault classification for single and mixed faults of a sensorless PMSM drive in dynamic operations using two external stray flux sensors alone. An order tracking method based on position estimation is proposed for resampling the measurement before generating features for machine learning algorithms. This eliminates the need for position sensors and makes the learning classifiers more robust to speed changes. The fault classifiers are trained using data from Profile 1, but they achieve high accuracy for detecting magnet defect or ITSC faults when tested on the datasets in Profiles 2 and Profile 3. This work also introduces a method of inducing local demagnetisation through heat treatments.

The comparative study shows that current and torque signals are not sensitive enough to detect the faults in the studied PMSM. However, data from two external stray flux sensors allow the classifiers to detect faults significantly better, although all learning classifiers are less robust to new speed profiles when trained for ITSC faults. FNN achieved the highest accuracy of the learning classifiers test in this study. It is recommended that the measurement period is set to a minimum of 10 rotor revolutions for computing one set of features without a significant loss of accuracy.

## Figures and Tables

**Figure 1 sensors-22-01216-f001:**
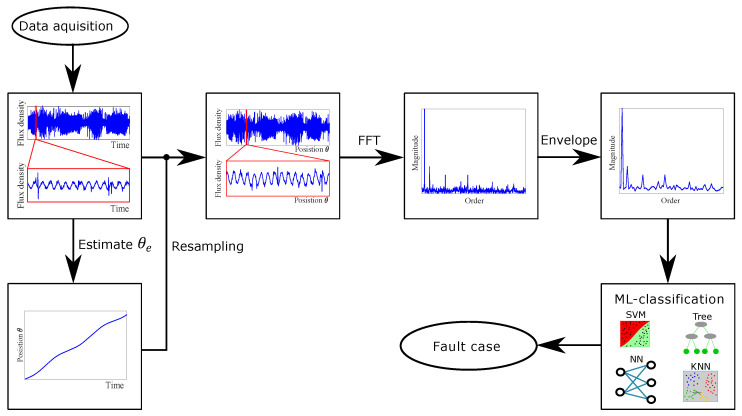
Block diagram of preparation of features.

**Figure 2 sensors-22-01216-f002:**
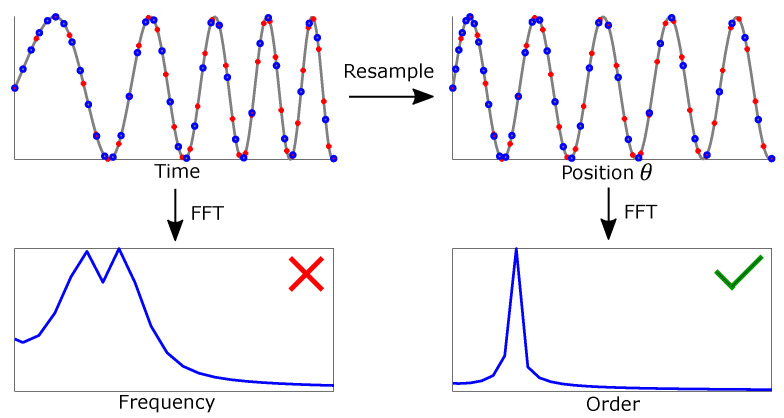
Visualising the benefit of the resampling process from time- to θ-domain.

**Figure 3 sensors-22-01216-f003:**
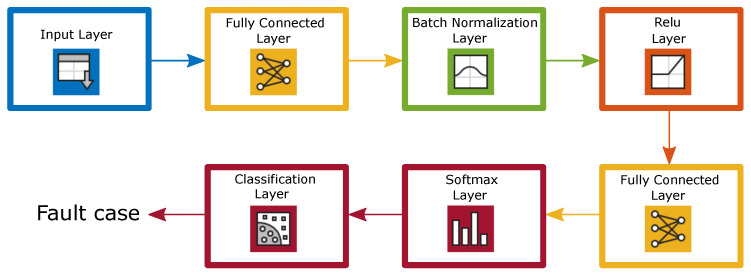
Block diagram of FNN used for fault classification.

**Figure 4 sensors-22-01216-f004:**
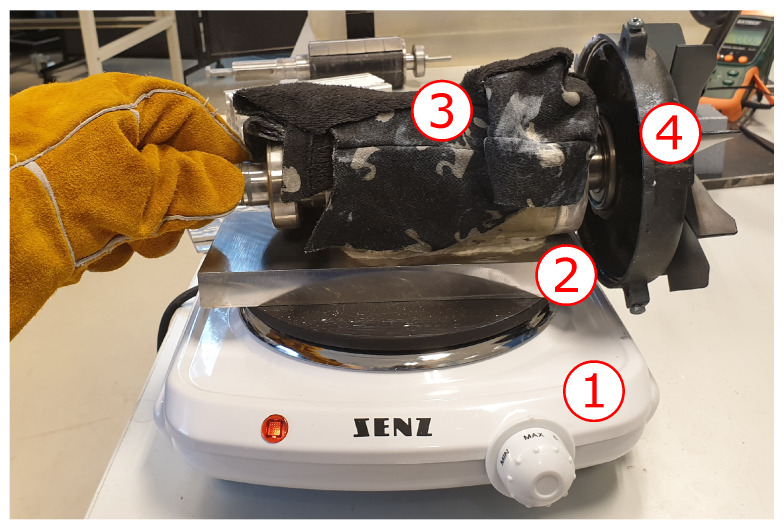
Thermal treatment setup consisting of (1) cooking plate, (2) solid aluminium block, (3) wet towel, and (4) the rotor.

**Figure 5 sensors-22-01216-f005:**
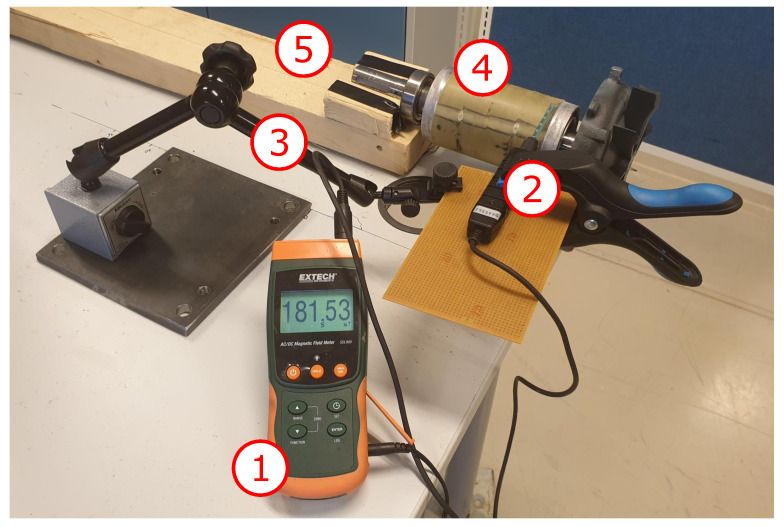
Setup for magnet strength measurement: (1) Extech magnetic AC/DC magnetic field meter, (2) measurement rod, (3) universal magnetic stand, (4) PM rotor, and (5) wooden frame.

**Figure 6 sensors-22-01216-f006:**
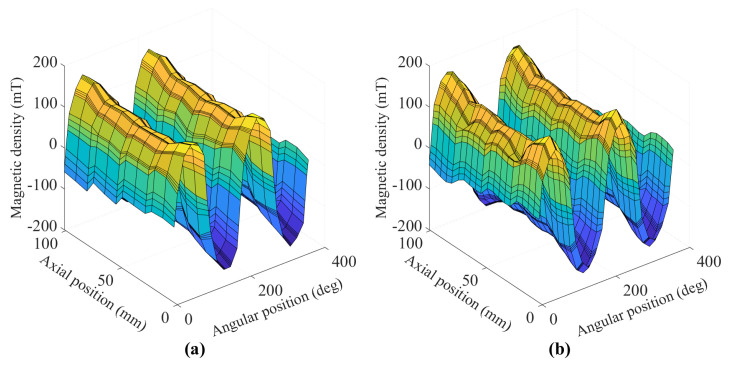
Side-by-side comparison of measured magnetic field strength of the permanent magnet rotor (**a**) before and (**b**) after the heat treatment.

**Figure 7 sensors-22-01216-f007:**
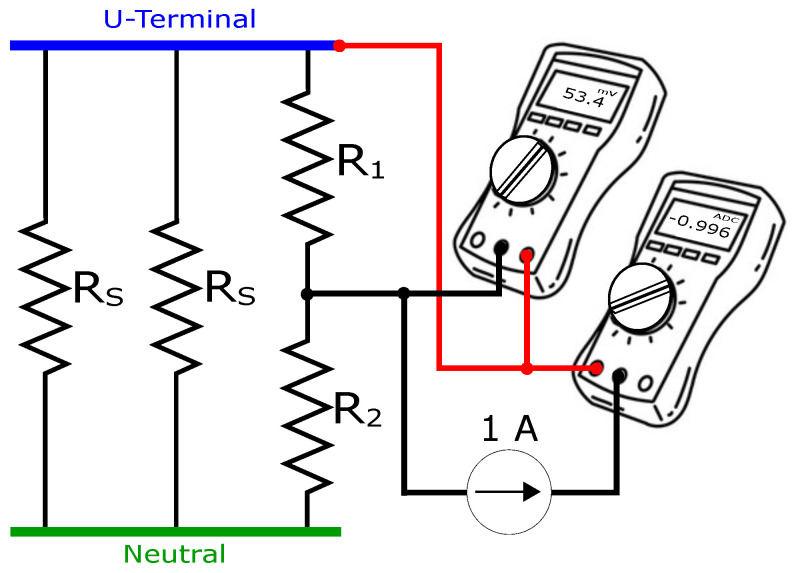
Sketch illustrating the setup for accurate resistance measurement.

**Figure 8 sensors-22-01216-f008:**
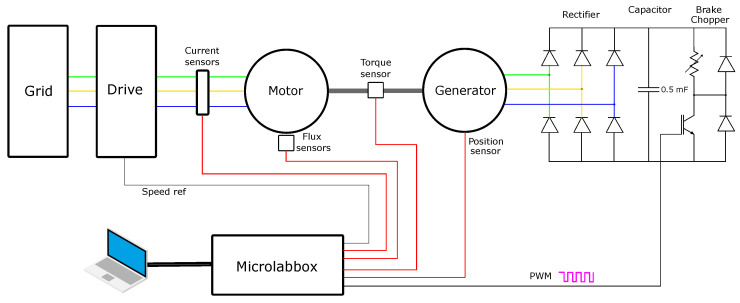
Schematic diagram of test bench.

**Figure 9 sensors-22-01216-f009:**
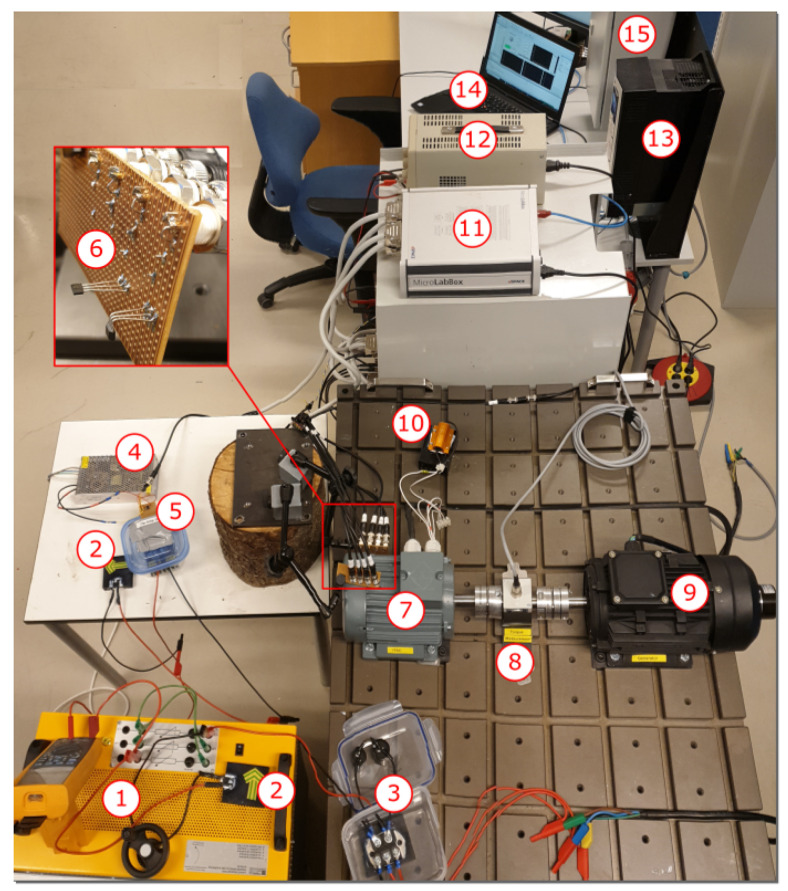
Overview of the test bench with (1) resistor bank, (2) flyback diode, (3) three-phase rectifier with capacitor bank, (4) 12 V DC supply, (5) IGBT brake chopper with op amp, (6) hall sensors, (7) PMSM, (8) torque sensor, (9) generator, (10) fault resistor, (11) Microlabbox, (12) 24 V DC-supply, (13) ABB drive, (14) office laptop, and (15) cabinet containing the current sensors.

**Figure 10 sensors-22-01216-f010:**
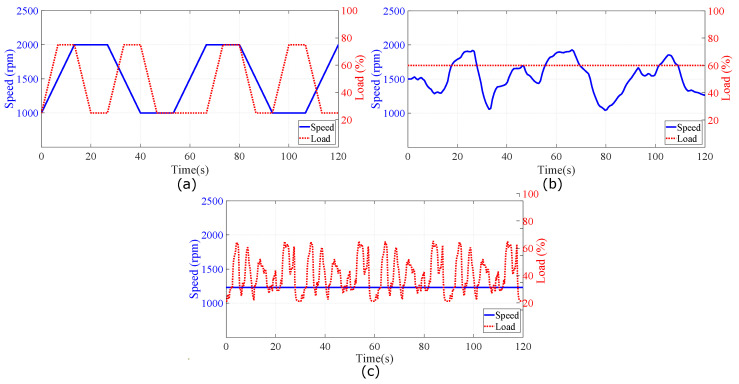
Visual representation of the three operation profiles: (**a**) Profile 1—Variable load and variable speed. (**b**) Profile 2—Constant load and variable speed and (**c**) Profile 3—Variable load and constant speed.

**Figure 11 sensors-22-01216-f011:**
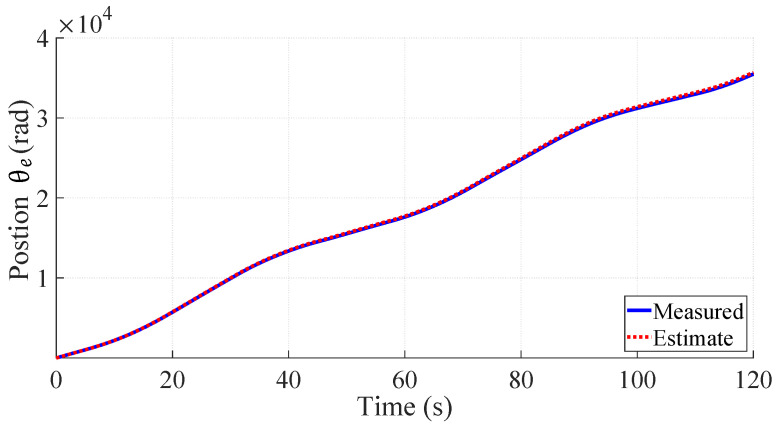
Comparing estimated $\theta_{e}$ with measured $\theta_{e}$.

**Figure 12 sensors-22-01216-f012:**
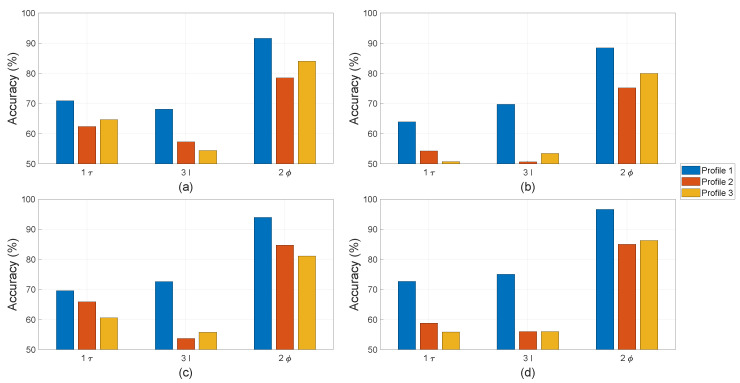
Detection accuracy with ITSC alone obtained by (**a**) EDT, (**b**) KNN, (**c**) SVM, and (**d**) FNN.

**Figure 13 sensors-22-01216-f013:**
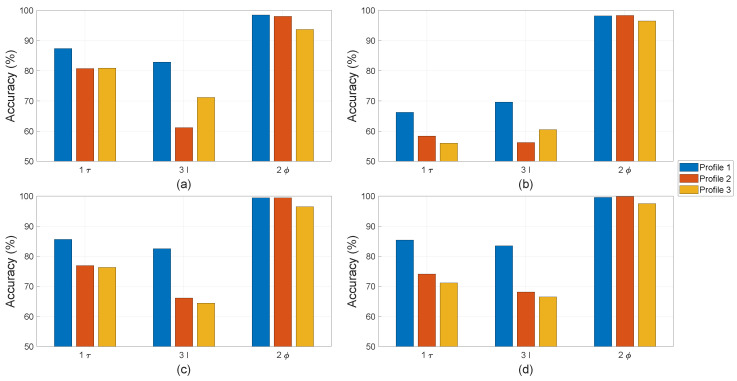
Detection accuracy with demagnetisation alone obtained by (**a**) EDT, (**b**) KNN, (**c**) SVM, and (**d**) FNN.

**Figure 14 sensors-22-01216-f014:**
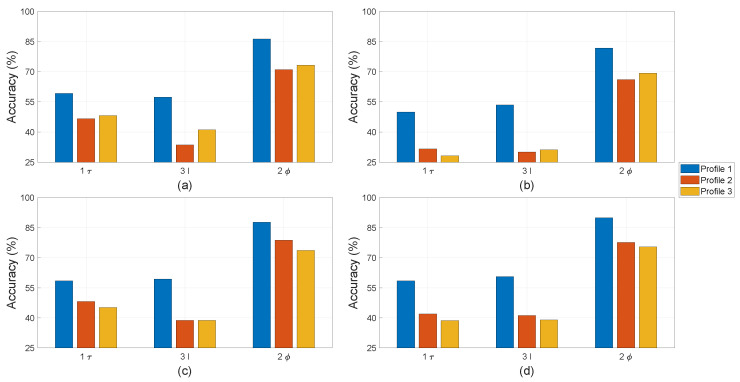
Classification accuracy with mixed fault obtained by (**a**) EDT, (**b**) KNN, (**c**) SVM, and (**d**) FNN.

**Figure 15 sensors-22-01216-f015:**
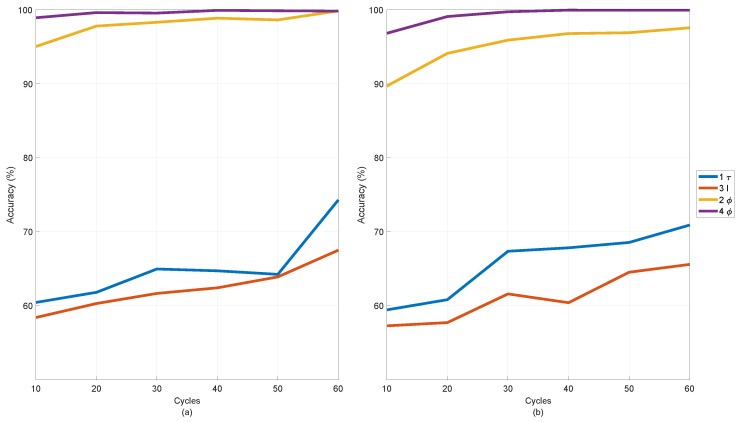
Performance of FNN-classifier for detecting local demagnetisation alone with spectrograms computed from different numbers of cycles in operations of (**a**) Profile 2 and (**b**) Profile 3.

**Table 1 sensors-22-01216-t001:** Nameplate parameters for IE5-PS2R 90 L.

Parameter	Value
Output power	3 kW
Nominal speed	3000 rpm
Number of poles	4
Nominal current	6 A
Nominal voltage	315 V
Phase resistance	0.6 Ω
Phase inductance	6 mH

**Table 2 sensors-22-01216-t002:** Training time of the machine learners.

Number of Classes	EDT	KNN	SVM	FNN
2	0.60 s	0.011 s	0.084 s	1.3 s
4	2.8 s	0.010 s	0.12 s	2.1 s

**Table 3 sensors-22-01216-t003:** Average classification accuracy (%) for detecting demagnetisation, ITSC, and mixed fault case.

Machine	Operation	Demagnetisation			ITSC			Mixed		
Learner	Profile	1 τ	3 *I*	2 ϕ	1 τ	3 *I*	2 ϕ	1 τ	3 *I*	2 ϕ
	1	85.6	82.5	**99.5**	69.6	72.6	**93.9**	58.5	59.4	87.8
SVM	2	76.9	66.1	**99.4**	65.9	53.7	84.7	48.2	38.8	78.7
	3	76.3	64.5	**96.4**	60.6	55.8	81.1	45.2	38.7	73.7
	1	66.2	69.6	**98.2**	63.9	69.7	88.5	50.0	53.5	81.8
KNN	2	58.4	56.2	**98.3**	54.3	50.6	75.2	31.6	30.1	66.1
	3	56.0	60.5	**96.5**	50.7	53.5	80.0	28.2	31.2	69.2
	1	87.4	82.9	**98.5**	70.9	68.2	**91.6**	59.2	57.4	86.3
EDT	2	80.8	61.2	**98.0**	62.3	57.3	78.5	46.6	33.6	71.1
	3	80.8	71.1	**93.7**	64.7	54.4	84.0	48.2	41.1	73.2
	1	85.4	83.5	**99.6**	72.7	75.1	**96.6**	58.5	60.6	89.9
FNN	2	74.1	68.1	**99.9**	58.8	56.0	85.0	42.0	41.2	77.7
	3	71.2	66.5	**97.5**	55.9	56.0	86.3	38.7	39.0	75.4

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
