# Peer review of "Mixed Fault Classification of Sensorless PMSM Drive in Dynamic Operations Based on External Stray Flux Sensors"

_sensors, 2022, doi:10.3390/s22031216_

Round 1

Reviewer 1 Report

In this work the authors present in very straightforward way a mixed fault classification methodology based on external stray flux sensors. In general the work is very well develop. Some comments:

1.- please develop more deeply section 2 specially the machine learning based methods.

2.- please add the nomenclature at the end of the manuscript.

Author Response

Please read the point-by-point response to your comments in the attached file 

Reviewer 2 Report

The paper carried out experiment and  worked on fault classification of permanent magnet synchronous motors. There are still some shortcomings as follows:

  1. You should list the hyperparameters of these algorithms.The prediction accuracy of decision trees should not be too different from algorithms such as KNN. It is suggested to use some ensemble algorithms to improve the prediction accuracy of decision trees.
  2. It is suggested to list the results of the four algorithms in a table, considering that the accuracy comparison across the bar charts is not obvious. 
  3. The radar chart used in Figure 17 is not as good as a normal line chart. You can use the X-axis to represent the number of cycles, and the Y-axis to represent the accuracy, and make different line charts according to different situations. It could better represent the information in the line charts.
  4. You have mentioned in section 1.2 on “To the author’s knowledge, most existing machine learning-based fault classifications are trained and tested on the same datasets or the same operational profiles, rendering a challenge of trustworthiness on other datasets or working conditions.” However, your specific method to solve the problem is not reflected in the paper.
  5. You can reduce the size of figures 9 to 12, and then add more explanations.
  6. For the fifth contribution, which is” Recommending minimum length of data sample for an accurate fault classification.”. You need have an instruction on it in detail.
  7. Compared with the experiment, the proposed algorithms are too simple. You can try more useful one, including Convolutional Neural Networks (CNN), ensemble learning (Random Forest, Boosting) instead of decision tree or feedforward neural network (FNN).

Author Response

Please read the point-by-point response to your comments in attached file
